# Quantifying Withanolides in Plasma: Pharmacokinetic Studies and Analytical Methods

**DOI:** 10.3390/nu16223836

**Published:** 2024-11-08

**Authors:** Alex B Speers, Axel Lozano-Ortiz, Amala Soumyanath

**Affiliations:** 1BENFRA Botanical Dietary Supplements Research Center, Oregon Health & Science University, Portland, OR 97239, USA; lozanoor@ohsu.edu (A.L.-O.); soumyana@ohsu.edu (A.S.); 2Department of Neurology, Oregon Health & Science University, Portland, OR 97239, USA; 3Department of Biology, Portland State University, Portland, OR 97201, USA

**Keywords:** *Withania somnifera*, ashwagandha, withanolides, withaferin A, liquid chromatography, mass spectrometry, pharmacokinetics

## Abstract

*Withania somnifera* (common name: ashwagandha; WS) is an Ayurvedic botanical that has become popular for its reputed effects on stress and insomnia. Research into the bioactive compounds responsible for the biological effects of WS has largely focused on withanolides, a group of steroidal lactones commonly found in the Solanaceae family. Until recently, however, it was unclear which, if any, withanolides were present in the plasma after the ingestion of WS products. The aim of this review is to summarize current knowledge regarding the plasma pharmacokinetics of withanolides found in WS and the analytical methods developed to detect them in plasma. Twenty studies (sixteen animal, four human) were identified in which isolated withanolides or withanolide-containing products were administered to animals or humans and quantified in plasma. Withanolides were commonly analyzed using reversed-phase liquid chromatography coupled to mass spectrometry. Plasma concentrations of withanolides varied significantly depending on the substance administered, withanolide dose, and route of administration. Plasma pharmacokinetics of withaferin A, withanolide A, withanolide B, withanoside IV, 12-deoxywithastramonolide, and withanone have been reported in rodents (C_max_ range: 5.6–8410 ng/mL), while withaferin A, withanolide A, 12-deoxywithastramonolide, and withanoside IV pharmacokinetic parameters have been described in humans (C_max_ range: 0.1–49.5 ng/mL).

## 1. Introduction

*Withania somnifera* (L.) Dunal (family Solanaceae) is a botanical with a long history of medicinal use in Ayurveda, where the roots, leaves, flowers, and seeds have all been used for a variety of ailments [1]. More commonly known as ashwagandha, *Withania somnifera* (WS) is categorized as an adaptogen, a group of natural compounds and plant extracts thought to “increase adaptability, resilience, and survival of organisms to stress” [2]. Pre-clinical studies and clinical trials have provided evidence to support its reputed adaptogenic effects, showing benefits for stress and stress-related disorders such as anxiety and insomnia in a variety of animal models and patient populations [3]. In part because of its status as an adaptogen, WS has garnered increased mainstream attention in recent years [4,5], along with growing popularity as a botanical dietary supplement [6,7].

In addition to its adaptogenic effects, another point of ongoing research interest is the safety of WS. WS products have generally been well tolerated in clinical trials; a systematic review of 30 clinical trials found no reports of serious adverse events or any significant changes to vitals and biochemical or hematological measures [8]. The only common side effects (>5%) reported in WS trials were somnolence, epigastric pain, and loose stools [8]. More concerning, however, are isolated case reports of liver toxicity [9,10]. In the majority of cases, liver tests normalized after discontinuation of WS [9,10]; however, there were three deaths in people who had underlying liver disease prior to taking WS [10]. There is evidence that WS modulates thyroid indices, suggesting caution in patients with thyroid disorder [11], and evidence for increased testosterone levels in men [12,13], contraindicating its use in individuals with prostate cancer. As with any botanical, there are also concerns about the potential for herb/drug interactions. There is mixed pre-clinical evidence for the effects of WS on cytochrome P450 isoenzymes, with some studies finding that WS induces CYP3A4 [14,15] and CYP1A2 [14] and inhibits CYP2B6 [15], while other studies found no effects [16,17,18].

Research into the active compounds responsible for the biological effects of WS has largely focused on withanolides, a group of naturally occurring steroidal lactones found in the plant families Solanaceae, Leguminosae, Labiatae, Myrtaceae, and Taccaceae [19], and particularly abundant in WS [20]. Chemical structures of the individual withanolides discussed in this review are shown in Figure 1. Withanolides have been extracted from the roots, leaves, and fruit of WS [21]; interestingly, the highest withanolide content is found in the leaves despite the root traditionally being the main part used in Ayurveda [21,22].

Withaferin A was the first withanolide discovered and isolated from WS in 1965 [23], and this compound has since gained attention for its anticancer activity against a wide range of cancer types in vitro (e.g., breast, ovarian, colon, leukemia, head and neck, multiple myeloma, neuroblastoma, and glioblastoma) via antioxidant, pro-apoptotic, anti-angiogenic, and anti-metastatic mechanisms [24]. In other disease models, withaferin A has demonstrated anti-inflammatory, anti-diabetic, neuroprotective, cardioprotective, and bone-building effects [24]. A growing number of withanolides have been isolated from WS in subsequent decades [21], several of which have also demonstrated therapeutic potential in pre-clinical models, including neuroprotective, anti-inflammatory, and anti-depressive effects (withanolide A [25,26,27,28,29], withanone [30,31,32,33,34], sominone [35,36,37], and 12-deoxywithastramonolide [38,39]).

While the pre-clinical data are promising for several withanolides, connecting these data to clinical outcomes for WS formulations is difficult without knowing which, if any, of these withanolides are present in the blood after ingestion and at what systemic concentrations they are biologically active. Robust pharmacokinetic (PK) data are needed to guide WS research, including mechanistic investigations, herb/drug interaction studies, and clinical dosing regimens (considering both safety and efficacy) for products containing WS. The aim of this review is to summarize (a) our current understanding of withanolide PKs and bioavailability and (b) the analytical methods used to quantify withanolides in plasma samples, with the goal of informing future studies on this widely used botanical.

## 2. Materials and Methods

Relevant studies were identified using the PubMed, Scopus, and Google Scholar databases, and from papers cited within the identified articles. Combinations of the following search terms were used: *Withania somnifera*, ashwagandha, Indian ginseng, poison gooseberry, winter cherry, pharmacokinetic, bioavailability, plasma, withanolide, withanoside, withaferin, 12-deoxywithastramonolide, withanone, sominone, C_max_, AUC, half-life, and T_max_. Studies in which a withanolide-containing product or isolated withanolide was administered to animals or humans and subsequently measured in the plasma were selected for review. Studies using multi-herbal preparations containing WS and withanolide-containing products not derived from WS (e.g., *Datura metel* L. flowers) were also included. Because the focus of this review is the quantification of withanolides in plasma, results from studies analyzing brain or tissue levels of withanolides are only discussed briefly. PK parameters are presented as they were reported in the publications, but concentrations have been converted to ng/mL when applicable to allow for easier comparisons between studies.

## 3. Results

### 3.1. Animal Studies

Sixteen animal studies were identified and are summarized in Table 1 and Table 2. Studies that administered isolated withanolides are listed in Table 1, while studies that administered withanolide-containing plant extracts are listed in Table 2. Withaferin A (eight studies) [40,41,42,43,44,45,46], withanolide A (one study) [47], or withanoside IV (one study) [37] were administered as isolated compounds in a total of ten studies. Withaferin A was administered to a variety of common mouse and rat strains at doses ranging from 0.5 to 70 mg/kg orally, 4.5 to 10 mg/kg intravenously, and 4 to 50 mg/kg intraperitoneally. The maximum plasma concentration (C_max_) range for withaferin A differed depending on the administered dose, route of administration, and animal model, ranging from 6 to 8410 ng/mL for oral administration, 29 to 3997 ng/mL for intravenous administration, and 7–10,000 ng/mL for intraperitoneal administration.

Across four studies (three in mice and one in rats) [41,42,43,45], the half-life of withaferin A fell within a relatively short time window (0.6 to 2.7 h). By contrast, in studies by Dai et al., 2019 [44] and Khedgikar et al., 2013 [48], the half-life of withaferin A after a 10 mg/kg oral dose was 7.6 and 7.1 h, respectively, and 4.5 h after a 5 mg/kg intravenous dose in Dai et al., 2019 [44]. Dai et al.’s study was also notable for reporting an oral bioavailability of 32.4% for withaferin A [44], significantly higher than Gupta et al., 2022, who administered a 70 mg/kg oral dose of withaferin A to female BALB/c mice and reported an oral bioavailability of 1.8% [41]. Two studies, one in mice [40] and one in rats [46], reported the detection of withaferin A in plasma but did not report PK parameters.

Withanolide A was administered to male Sprague Dawley rats at a dose of 25 mg/kg orally and 2 mg/kg intravenously [47], resulting in plasma C_max_ values of 48 and 86 ng/mL and half-lives of 2.23 and 2.21 h, respectively. The oral bioavailability of withanolide A in this study was 5.2% [47]. Withanoside IV, a withanolide glycoside isolated from a methanolic WS root extract, was administered to male ddY mice at 1000 µmol/kg (782.9 mg/kg) orally [37]. Withanoside IV was not detected in the plasma at any time point; however, its aglycone, sominone, was detected by high-performance liquid chromatography coupled to ultraviolet detection (HPLC-UV) and liquid chromatography coupled to mass spectrometry (LC-MS) at 3 h and reached maximal concentration at 7 h, though absolute concentrations were not reported.

Six studies administered withanolide-containing plant extracts, which included aqueous [53], alcoholic [49], and hydroalcoholic [52] WS root extracts, a withanolide-rich fraction from a hydroalcoholic WS root extract [50], a withanolide aglycone-enriched fraction from a hydroalcoholic WS root extract [52], a polyherbal hydroalcoholic extract containing five botanicals including WS [51], and a hydroalcoholic flower extract from *Datura metel* L. [54], another botanical in the Solanaceae family. All the withanolide-containing plant extracts were administered orally, at a dose range of 50 to 5000 mg/kg, to various rat strains (three studies), Swiss albino mice (two studies), and guinea pigs (one study). The withanolides measured in these six studies included withaferin A (five studies), withanolide A (three studies), withanolide B (three studies), 12-deoxywithastramonolide (three studies), withanone (two studies), withanoside IV (one study), and withanoside V (one study). When reported, the concentrations of individual withanolides in the administered doses were low, ranging from 0.002% to 2.680% of the total extract. The one exception was the study by Srivastava et al., 2013 [52], where a withanolide aglycone-enriched fraction administered to male Swiss albino mice contained 70.56% 12-deoxywithastramonolide and 22.94% withanolide A. All withanolides examined were detected in the plasma in these six studies, though withanoside V, withanolide B, and withanone were below the lower limit of quantitation (LLOQ; 3 ng/mL) in a study by Modi et al., 2022 [49], in which male Sprague Dawley rats were administered 500 mg/kg of an alcoholic WS root extract.

Four of the included animal studies also reported tissue concentrations of withanolides. Wang et al., 2019 [45] administered 4.5 mg/kg oral withaferin A to Sprague Dawley rats and measured its concentration in tissues over a 5 h period. Peak concentrations measured in homogenates (1 g tissue/mL) of various organs were as follows: stomach (C_max_ = 2672 ng/mL), intestine (C_max_ = 1330 ng/mL), heart (C_max_ = 1808 ng/mL), liver (C_max_ = 14.6 ng/mL), lung (C_max_ = 1805 ng/mL), kidney (C_max_ = 1710 ng/mL), and spleen (C_max_ = 980 ng/mL). Gambhir et al., 2015 [40] administered 50 mg/kg withaferin A intraperitoneally to Swiss albino mice and measured tissue distribution over 4 h. Withaferin A was not detected in the brain but reached maximum levels at 4 h in plasma (~10 µg/mL), liver (~10 µg/g), and intestine (~15 µg/g) (specific concentrations not reported). Dadge et al., 2023 [50] orally administered 50 mg/kg of a withanolide-rich fraction from a hydroalcoholic root extract to male Sprague Dawley rats. All five measured withanolides were detected in brain homogenates prepared from tissue collected over 8 h, reaching peak levels of between 3.5 and 6 ng/g in the first hour. Withaferin A showed sustained peak levels for up to 2 h, whereas the other four withanolides (withanolide A and B, withanone, and 12-deoxywithastramonolide) declined rapidly after reaching a peak value. A fourth study by Mukherjee et al., 2019 [55] administered 10 mg/kg withanolide A intranasally to Swiss albino mice and measured concentrations of the compound in different brain regions. Withanolide A was detected in methanolic homogenates (tissue–methanol ratio not specified) of the cortex (C_max_: 11.9 ± 1.3 µg/mL, T_max_: 1 h, T_1/2_: 2.6 ± 0.4 h) and cerebellum (C_max_: 17.3 ± 2.4 µg/mL, T_max_: 1 h, T_1/2_: 3.0 ± 0.6 h).

### 3.2. Human Studies

The quantification of withanolides in human plasma has been reported in four studies (Table 3). These studies administered a variety of WS botanical dietary supplements: AshwaMAX, a WS root extract standardized to 4.5% *w*/*w* withaferin A; Prolanza™, a WS root extract; KSM-66^®^, an aqueous WS root extract standardized to 15 mg withanolides; WS-35, a hydroalcoholic WS root and leaf extract, standardized to 40% *w*/*w* total withanolides comprising 35% *w*/*w* withanolide glycosides; WS-2.5, a WS extract (plant part not reported), standardized to 2.5% withanolides; and Witholytin^®^, a WS root extract, standardized to ≥1.5% *w*/*w* total withanolides. WS supplements were administered to healthy adult males (three studies) and patients with advanced stage high-grade osteosarcoma (one study).

The withanolide content per dose varied depending on the WS supplement being administered. In studies where two different WS botanical dietary supplements were compared, products were matched based on withanolide content per dose. For example, each dose of Prolanza™ and KSM-66^®^ contained 30 mg withanolides [57], while each dose of WS-35 and WS-2.5 contained 185 mg withanolides [58]. The withanolide content of Witholytin^®^ was reported to be 7.97 mg withanolides per dose [59]. AshwaMAX is standardized to withaferin A rather than total withanolides; participants received between 72 and 216 mg withaferin A per dose [56].

Analyzed withanolides included withaferin A [56,58,59], withanolide A [57,58,59], 12-deoxywithastramonolide [57,59], withanoside IV [58,59], withanoside V [59], and sominone [59]. The highest reported plasma C_max_ value for an individual withanolide was 49.50 ng/mL of withaferin A following oral administration of 480 mg WS-35 [58]. Withaferin A was also detected after oral administration of WS-2.5 (C_max_ = 10.79 ng/mL) and Witholytin^®^ (C_max_ = 2.88 ng/mL), but not AshwaMAX [56], where a less sensitive detection method of ultraviolet spectroscopy (LLOQ 50 ng/mL) was used for the analysis. Withanolide A (C_max_ range: 0.09 to 4.74 ng/mL), 12-deoxywithastramonolide (C_max_ range: 0.61 to 5.50 ng/mL), and withanoside IV (C_max_ range: 0.64 to 7.23 ng/mL) were all detected in plasma when measured. In the Witholytin^®^ study [59], withanoside V and sominone were detected in plasma, but were below the LLOQ (0.25 ng/mL).

### 3.3. Analytical Methods to Detect Withanolides in Plasma

Techniques used to analyze withanolide content in plasma in the previously discussed studies are summarized in Table 4 (animal studies) and Table 5 (human studies). By far, the most common method reported for the separation and quantitation of plasma withanolides was liquid chromatography coupled to tandem mass spectrometry (LC-MS/MS). In some cases, ultraviolet (UV) detection was used alongside, or instead of, mass spectrometry [37,40,48,56].

All the reported studies used reversed-phase LC with either isocratic or gradient elution to separate the withanolides of interest. The stationary phase in all cases consisted of C18 material, although the brand, particle size, and column dimensions varied between studies. The mobile phase consisted of mixtures of water with acetonitrile and/or methanol as an organic modifier, often with pH modification using acids (formic acid or phosphoric acid), bases (triethylamine), salts (ammonium formate or ammonium acetate), or buffers (potassium dihydrogen phosphate).

When UV detection was employed, the detection wavelengths were 290 nm [40], 220 nm [37], 225 nm [56], and 370 nm [48]. For the MS/MS detection of withanolides, most studies employed positive ionization methods. However, two studies also used negative ionization [51,58]. Addition of ammonium salts to the mobile phase [41,44,47,53,58,59] allows for the generation and detection of withanolides as their ammonium [M+18]+ adducts as well as, or instead of, as their protonated forms [M+1]+. However, even in the absence of added ammonium salts in the mobile phase, detection of [M+18]+ forms as a precursor ion has been reported for withanone [50], withanolide A [45,49], and withanosides IV and V [49]. While most studies used molecular ions or their ammonium adducts of withanolide analytes as precursor ions for MS/MS reactions, one study [49] used the MS/MS transition of a larger to smaller fragment (417.25/263.15) for the detection of withanone (monoisotopic mass: 470.3). In the study by Kim et al., 2023 [58], a formate adduct [M+45]+ of withanoside IV (monoisotopic mass 782.4) was used as the precursor ion, for the MS/MS transition 827.4404/763.3592.

A variety of methods have been used for plasma sample clean up (i.e., removal of interferents such as proteins, lipids, and salts) prior to LC analysis. The simplest method applied was protein precipitation (or protein “crash”) achieved by the addition of an organic solvent such as acetonitrile or methanol to the plasma sample, followed by centrifugation to precipitate proteins, and LC-MS analysis of the supernatant sometimes following a concentration step [42,43,51,54,56]. In other cases, withanolides were extracted from the plasma by solvent extraction either directly [45,53] or following protein precipitation [41,44,47,50]. Tert-butyl methyl ether (TBME) [47,50,53] or ethylacetate [41,44,45] were used as extraction solvents. Solid phase extraction (SPE) of the withanolides using Bond Elute C18 [49,59], Bond Elute PLEXA [57], or Oasis HLB [37,58] cartridges were employed in some studies. Specific SPE protocols varied by study and interested readers are recommended to consult the original articles for details. Broadly, plasma samples were diluted with water [59] or aqueous acid [58] prior to loading onto preconditioned SPE cartridges. Impurities were washed off with water, and withanolides were eluted using methanol [49,57,58,59].

Just over half of the studies [41,44,45,49,50,53,54,55,58,59] reported standard validation parameters (linearity, accuracy, precision, limits of detection, and quantitation) for the LC-MS/MS methods employed. In validation experiments, the LLOQ for withanolides in plasma has generally been 3 ng/mL or less. For withaferin A, the reported LLOQ ranged from 0.2 ng/mL [44] to 3 ng/mL [49], with several studies reporting intermediate values within this range [41,45,50,53,58,59]. For withanolide A, LLOQ values similarly ranged [49,50,53,58,59] from 0.25 ng/mL [59] to 3 ng/mL [49]. For withanolide B, LLOQ values of 2 ng/mL [50], 3 ng/mL [49], and 6 ng/mL [54] were noted. Withanosides IV and V and 12-deoxywithastramonolide were all quantifiable in plasma with LLOQ values of either 0.25 ng/mL [59] or 3 ng/mL [49], with additional studies reporting LLOQ values for withanoside IV [58] and 12-deoxywithastramonolide [50] of 1 ng/mL and 2 ng/mL, respectively. Finally, two studies measuring withanone reported LLOQ values of 1 ng/mL [50] and 3 ng/mL [49].

For the analysis of withanolides in brain and other tissues, tissue homogenates were initially made. Gambhir et al., 2015 and Dadge et al., 2023 did not specify the homogenization solvent [40,50], but Wang et al., 2019 used saline solution containing ascorbic acid for homogenization [45]. Gambhir et al., 2015 [40] did not describe sample preparation or instrumental details for the HPLC analysis data reported. Dadge et al., 2023 and Wang et al., 2019 processed brain and other tissue homogenates using the same clean up methods used for plasma in their respective studies (Table 4) [45,50] and have validated their LC-MS/MS analytical methods for both plasma and tissues. Mukherjee et al., 2019 only reported withanolide A levels in mouse brain regions (not plasma). In that study [55], cortex and cerebellum regions were excised and homogenized in methanol, followed by centrifugation and filtration (0.22 mm) to obtain a cell-free methanol extract for HPLC-UV analysis.

## 4. Discussion

WS extracts have gained popularity among consumers as botanical dietary supplements [60] and have shown promising effects in early clinical trials [3]. However, there are still significant gaps in our understanding of how withanolides, the purported active compounds in WS, contribute to the plant’s biological effects. Withanolides are a large group of naturally occurring steroidal lactones, of which over 40 have been extracted from the roots, berries, and aerial parts of WS [49]. A small number of withanolides have been studied as isolated compounds, demonstrating therapeutic potential in a variety of pre-clinical models [24,25,26,27,28,29,30,31,32,33,34,35,36,37,38,39].

Understanding which withanolides are bioavailable and the levels they attain in plasma and target tissues is an important part of exploring their role as active compounds of WS. Such studies also help to bridge the gap between in vitro and in vivo studies, allowing the in vivo relevance of activities observed at specific concentrations in vitro to be assessed. Mechanistic research can then focus on those compounds appropriate to the in vivo situation. PK and bioavailability studies of withanolides are also translationally relevant. They can be used to guide human dosing regimens for WS products, evaluate the clinical risk of potential toxic reactions or herb/drug interactions observed in vitro, and provide insight into the differential effects of various WS products.

To support ongoing research into evaluating the PKs and bioavailability of this important group of phytochemicals, the goal of this review was to (a) review the pre-clinical and clinical studies that have quantified individual withanolides in animal and human plasma after the administration of isolated withanolides or withanolide-containing plant extracts and (b) summarize the analytical methods that have been used to detect and quantify withanolides in plasma.

Withanolides are found in both aglycone and glycosidic forms (Figure 1). There are significantly more published data on the PKs and bioavailability of withanolide aglycones than on withanolide glycosides. Many naturally occurring glycosidic compounds undergo hydrolysis in the gastrointestinal tract, such that only aglycone forms are observed in the plasma. For example, following oral administration of an extract of *Centella asiatica* containing the triterpene glycosides asiaticoside and madecassoside to humans, only the respective aglycone forms, asiatic acid and madecassic acid, were detected in plasma [61,62]. In the case of WS, the aglycone sominone, but not the parent glycoside withanoside IV, was detected in mouse plasma following oral administration of withanoside IV [37]. However, both withanoside IV (C_max_ = 13.83 ng/mL) and withanoside V (<LLOQ) were detected in plasma of rats orally administered a WS alcoholic extract [49]. Withanoside IV has also been quantified in human plasma PK studies (C_max_ = 0.64–7.23 ng/mL) following the oral administration of three different WS products [58,59]. Withanoside V was detected (<LLOQ) in plasma but not quantified in one study [59]. Oral bioavailability was not quantified in any of these studies, but these data demonstrate that withanolide glycosides can be absorbed following oral administration. This is not surprising since the structurally related cardiac glycoside drug, digoxin, is systemically bioavailable following oral administration [63].

In three WS products tested in humans [58,59], the absorption rate for withanoside IV was very similar (T_max_ = 1.57–1.76 h) and was within the range seen for withanolide aglycones measured from the same products (T_max_ = 0.9–2.28 h). The half-lives (T_1/2_) for withanoside IV were 8.86 h (WS-35), 2.43 h (WS-2.5), and 4.41 h (Witholytin^®^). In each of these products, the aglycones measured (withaferin A, withanolide A, 12-deoxywithastramonolide) had similar or longer half-lives than the glycoside withanoside IV, ranging from 10.35 to 10.95 h (WS-35), 1.88 to 4.03 h (WS-2.5), and 2.73 to 4.19 h (Witholytin^®^). Differences in T_max_ and T_1/2_ seen for the different aglycones measured within a given study could either reflect true differences in their individual rates of absorption and elimination or may also be due to differences in their generation in the gastrointestinal tract from parent glycosides, thus delaying or prolonging their absorption and elimination.

As suggested in the examples above, plasma PK parameters seen for individual compounds varied between studies. As would be expected, dose, frequency, and route of administration are important factors in this variation. A clear dose-related increase in C_max_ (6.05, 14.60, and 21.80 ng/mL) was observed when withaferin A was administered orally at increasing single doses (0.5, 1.5, and 4.5 mg/kg) to rats [45]. When administered intravenously, 4.5 mg/kg withaferin A resulted in a higher C_max_ (29.10 ng/mL) and had a shorter T_max_ (0.33 h) than when given orally (T_max_ = 0.86 h) [45]. Similarly, when withaferin A was administered at either 4 or 8 mg/kg intraperitoneally per day to rats for 10 weeks, plasma levels measured at week 10 were 65.93 and 138.1 ng/mL, respectively [46]. Comparing these two independent studies both performed in rats, the use of the intraperitoneal route of administration and a multiple dosing schedule were associated with significantly higher plasma levels of withaferin A than single dosing.

However, independent studies administering withaferin A at similar doses and routes to the same species have also reported widely differing peak plasma levels. An example of a discrepancy between studies in the same species was seen following intraperitoneal administration of withaferin A in mice. Thaiparambil et al., 2011 reported a C_max_ for withaferin A of 847.1 ng/mL following a 4 mg/kg intraperitoneal dose [43], whereas Patel et al., 2019 reported a C_max_ for withaferin A of 6.7 ng/mL following a 5 mg/kg intraperitoneal dose [42]. Different rodent strains and sexes were used in the two studies (male C57BL/6N mice versus female BALB/c mice, respectively) along with different vehicles. Thaiparambil et al., 2011 [43] injected withaferin A with a vehicle of 10% dimethyl sulfoxide (DMSO), 40% Cremophor-EL, and 50% PBS, leading to rapid absorption (T_max_ = 5 min), while Patel et al., 2019 [42] dissolved withaferin A in DMSO and diluted it with saline prior to injection, which may have resulted in slower absorption (T_max_ = 20 min). Methodological details for withaferin A LC-MS analysis were not provided for these studies and cannot therefore be compared as a potential reason for this wide difference. Interestingly, a study in which withaferin A was administered at a dose of 50 mg/kg intraperitoneally to male mice [40] resulted in a withaferin A plasma concentration of ~10 µg/mL (determined by LC-UV), which corresponds proportionately to the C_max_ value (approximately 0.8 µg/mL) obtained by Thaiparambil [43] for a 4 mg/kg intraperitoneal dose. Both of these studies [40,43] were conducted in male mice, raising the possibility of significant sex differences in withaferin A disposition accounting for the low C_max_ in female mice [42].

A similar discrepancy in peak plasma levels was observed following oral and intravenous administration of withaferin A. Dai et al., 2019 reported a C_max_ for withaferin A of 619 ng/mL in male rats following a 10 mg/kg oral dose and an oral bioavailability of 32.4% [44]. Wang et al., 2019 [45] reported a much lower C_max_ value of 21.8 ng/mL after intra-gastric administration of 4.5 mg/kg withaferin A to Sprague Dawley rats at about half the dose of Dai et al., 2019 [44]. In the same two studies, intravenous doses of 5 mg/kg [44] and 4.5 mg/kg [45] resulted in mean C_max_ values of 3048 ng/mL and 29.1 ng/mL, respectively. The reasons for these large differences are unclear as vehicle information is not provided for the study by Wang et al., 2019 [45], and both studies used very similar, validated methods for plasma work up and withaferin A analysis (Table 4). Sex-related differences could again offer a possible explanation for the observed differences as Dai et al., 2019 [44] used male rats, while Wang et al., 2019 [45] did not report the sex of the rats. However, Khedgikar et al., 2013 present contradictory data to the hypothesis of a sex-related difference [48]. When female BALB/c mice were administered an oral dose of 10 mg/kg withaferin A (the same dose administered in Dai et al., 2019 [44]), the reported C_max_ was 8410 ng/mL [48], significantly higher than any other oral withaferin A study.

While the relevant methodological details are not always reported, the vehicle in which the test material is delivered may influence PK parameters. As indicated earlier, Khedgikar et al., 2013 [48] and Dai et al., 2019 [44] reported a C_max_ for withaferin A of 8410 ng/mL and 619 ng/mL following a 10 mg/kg oral dose; Dai et al., 2019 also reported an oral bioavailability of 32.4% [44]. In comparison, the next highest reported C_max_ for withaferin A was 142 ng/mL following a 70 mg/kg dose, with an oral bioavailability of only 1.8% [41]. The significant differences in the C_max_ and oral bioavailability of withaferin A may be influenced by the vehicles used in the respective studies [41]. In Khedgikar et al., 2013, withaferin A was dispersed in gum acacia in a 1:1 ratio; gum acacia is a well-known emulsifier that has previously been shown to increase intestinal absorption of omega-3 fatty acids by 321% [64]. Dai et al., 2019 [44] dissolved withaferin A in an ethanol–solutol HS 15-distilled water (10:5:85, *v*:*v*:*v*) mixture, which may have led to rapid oral absorption (T_max_ = 6.6 min), whereas Gupta et al., 2022 [41] administered withaferin A in 100 µL of 0.5% carboxymethylcellulose, which creates a drug suspension possibly resulting in slower absorption (T_max_ = 30 min) [41]. Gupta et al., 2022 hypothesized that the rapid absorption of withaferin A observed in Dai et al., 2019 may also have saturated the CYP enzymes, leading to decreased metabolism and a dramatically higher C_max_ and AUC [41].

It is also interesting to compare the PKs of compounds when administered alone versus as part of a mixture of compounds extracted from WS. Existing data only allow these comparisons to be made across independent studies. Comparing the PK parameters of similar doses of orally administered withaferin A given to rats either as an isolated compound at 4.5 mg/kg [45] or as part of a complex extract at 4.84 mg/kg [49], the isolated compound resulted in a considerably lower C_max_ (21.80 ng/mL vs. 124.42 ng/mL), with a later T_max_ (0.86 h vs. 0.25 h) and a shorter half-life (1.15 vs. 3.15 h) compared to the extract. These results may indicate that other components in the extract strongly influence the disposition of withaferin A by influencing transporter systems or metabolizing enzymes. Alternatively, the differences may be due to the presence of precursor compounds (e.g., glycosides) in the extract that give rise in vivo to additional withaferin A, and over a more sustained period after ingestion.

Different extract compositions may also affect the PK parameters of a given constituent. A WS extract delivering a small oral dose of withaferin A (0.0015 mg/kg) resulted in a disproportionately high C_max_ of 6.50 ng/mL [50] in rats compared to the C_max_ (124.42 ng/mL) following oral administration of a different WS extract in rats, equivalent to 4.84 mg/kg withaferin A [49]. Two extracts were compared in a single study by Srivastava et al., [52]. PK parameters for 12-deoxywithastramonolide and withaferin A were examined in mice that received 200 mg/kg of either a hydroalcoholic extract of WS root (WSC) or an aglycone-enriched fraction (WSAg) of the WSC extract, both administered orally. The doses of WSC and WSAg delivered 5.36 and 141.12 mg/kg, respectively, of 12-deoxywithastramonolide. However, the corresponding Cmax values for this analyte of 120 ng/mL (WSC) and 50 ng/mL (WSAg) indicated a disproportionately high value for the WSC extract. In both studies [50,52], the unexpectedly high values could be due to the presence of glycoside precursors converting to the analyte in vivo.

Interestingly, however, a different result was seen when comparing withaferin A plasma levels. Almost identical oral doses (2.26 and 2.3 mg/kg) of withaferin A delivered as WSC and WSAg (200 mg/kg) displayed C_max_ values of 30 ng/mL and 60 ng/mL, respectively) [52]. The extract giving the higher C_max_ value was said to be more enriched in aglycones, reducing the likelihood of glycosides generating withaferin A in vivo. In this case, a less selective analytical method (LC-UV) was used and was not described in detail in the original article. Therefore, the possibility that other aglycones interfered with the assay results, for both withaferin A and 12-deoxywithastramonolide, cannot be excluded.

As in the animal studies, human plasma levels of withanolides varied with the dose and type of WS material administered. In humans, the highest plasma withanolide concentrations were seen for WS-35 and WS-2.5, both of which had the highest withanolide content per dose (185 mg) [58]; however, Witholytin^®^ (7.97 mg withanolides per dose) produced much higher plasma concentrations of withanolide A and 12-deoxywithastramonolide compared to Prolanza™ and KSM-66^®^, each containing 30 mg of withanolides per dose [57,59]. There are several possible reasons for these apparent inconsistencies. While all three studies reported total withanolide content per dose, individual withanolide content was not reported, apart from withanoside IV and withanoside V in Vaidya et al., 2023 [59]. Without knowing the individual withanolide content for each product, it is difficult to compare PK outcomes between products, as individual withanolide content could vary greatly depending on the extraction method used to make the product. Other natural WS components present in the products (which would vary with extraction method) may affect PK profiles of the withanolides of interest by competing for transport mechanisms or metabolizing enzymes. Individual formulations prepared by manufacturers may also influence the release characteristics of the constituent withanolides. For example, one product studied, Prolanza^TM^ [57], is described as a sustained release preparation and had a much longer half-life for withanolide A and 12-deoxywithastramonolide (approximately 7.5 h) compared to these same compounds in KSM-66 (1–2 h) or Witholytin^®^ (2.73–4.19 h) [59]. Differential release patterns of the withanolides in the GI tract could also affect their gut microbial transformation, resulting in variations in the bioavailability of similar administered doses. Finally, the study population’s characteristics will affect the PK parameters observed, although three of the four human studies reviewed here were performed in healthy adult males of a similar age.

Botanical dietary supplements containing WS generally contain complex extracts. The influence of additional components on the plasma PK parameters of any individual withanolide make it difficult to extrapolate data between extracts. This also highlights the need for careful and comprehensive characterization and reporting of the components of complex botanical extracts being examined in PK or other scientific studies.

Given the observed variability in individual withanolides’ PK parameters based on the material being administered, it would be optimal to be able to associate plasma levels of specific withanolides with their therapeutic activities and adverse effects, as this would allow safe and effective doses of a given formulation to be identified. It would also facilitate interspecies translation of experimental results

In our literature search, only one study was found where both therapeutic activities and associated plasma withanolide levels were reported for individual withanolides. Khedgikar et al., 2013 administered 1, 5, or 10 mg/kg/day oral withaferin A to osteopenic ovariectomized female mice over 8 weeks, and conducted a PK study in these same mice using a single dose of 10 mg/kg withaferin A [48]. At a daily dose of 10 mg/kg for 8 weeks, withaferin A demonstrated an anabolic effect on osteoporotic bone, causing osteoblast growth and differentiation [48]. The steady-state plasma levels of withaferin A achieved after 8 weeks would likely have been even higher than the C_max_ of 8410 ng/mL following single dosing. This C_max_ is significantly higher than any seen in the other withaferin A studies discussed in this review, regardless of species and route of administration (with one exception [40]). As noted earlier, the vehicle used (1:1 gum acacia) may have contributed to the high plasma levels. Given the lesser effects seen at 1 and 5 mg/kg withaferin A [48], it is unlikely the beneficial effects of withaferin A on osteoporotic bone would be observed with other withaferin A or WS preparations described in this review.

More commonly, the safety and tolerability of ashwagandha preparations have been evaluated alongside the PK parameters of constituent withanolides. In three clinical studies [57,58,59] where plasma C_max_ values of individual withanolides ranged from approximately 1 to 50 ng/mL, no adverse effects were reported following single doses of complex WS extracts. A fourth clinical study [56] examined the adverse effects of four escalating dose levels of a complex ashwagandha product each administered for 30 days, a paradigm more relevant to regular use of supplements. In this study, only mild (grade 1) or moderate (grade 2) adverse events (fatigue, fever, rash, diarrhea, edema, and abnormal liver function tests) were observed. Unfortunately, plasma withaferin A levels were below the 50 ng/mL detection limit of the analytical method used in this study and cannot be associated with the effects seen.

Toxicology data for individual withanolides are mostly limited to withaferin A. In a mouse model of breast cancer, an intraperitoneal dose of up to 4.0 mg/kg (plasma C_max_ = 847.1 ng/mL) caused no significant change in liver or lung histology [43]. An intraperitoneal dose of 5 mg/kg withaferin A (plasma C_max_ = 6.7 ng/mL) caused no histologic changes in the liver and did not alter AST or ALT in a mouse model [42]. At higher intraperitoneal doses (10 and 40 mg/kg every day for six days), however, withaferin A caused a relative increase in thrombocytes and leukopenia, respectively, in Ehrlich ascites tumor-bearing mice [65]. PK parameters were not reported in this study. For oral administration, an in silico study predicted that two-thirds of the 75 withanolides evaluated would have an LD_50_ value under 100 mg/kg in rats following oral dosing, with lower LD_50_ values (<50 mg/kg) for withanolide glycosides compared to aglycones [66]. The predicted LD_50_ for withaferin A in this study was 96.5 mg/kg [66]. However, in mice, withaferin A administered as a single oral dose was found to be well tolerated with no significant drug-related toxicities up to 2000 mg/kg [41]. When administered over 28 days, a No-Observed Adverse Effect Level (NOAEL) of at least 500 mg/kg oral withaferin A was reported in mice [41]. Unfortunately, plasma levels of withaferin A were not reported for the highest tolerated doses mentioned above.

While most toxicity studies have focused on withaferin A, the withanolide withanone has been identified as a potential contributing factor to WS-associated liver toxicity due to its ability to cause DNA damage when detoxification by glutathione is limited in vitro [67]. However, a plasma C_max_ value for withanone following single doses of a WS product has only been reported in one rat study [50] and was not associated with a biological effect.

These examples illustrate the need for the concomitant evaluation of plasma levels of withanolides when studying in vivo biological effects and toxicity. Ideally, these studies would be conducted with single compounds. It is not possible to make direct associations between the plasma levels of individual compounds and biological effects when complex botanical mixtures are administered as more than one compound may be active. Conversely, if a complex mixture is well tolerated or lacks a particular biological effect in vivo, the associated plasma levels of individual withanolides can also be assumed to be so (absent of any interactions).

In common with many bioanalytical applications, liquid chromatography coupled to tandem mass spectrometry (LC-MS/MS) has emerged as the method of choice for the quantification of withanolides in both animal and human plasma. Generally, reversed-phase LC and positive ionization MS have been employed and several common methods (protein precipitation, solvent extraction, solid phase extraction) have been used for plasma clean up prior to analysis. The greatest advantages of LC-MS/MS are that it provides greater sensitivity and selectivity compared to other common detection methods such as LC coupled to UV detection. For example, a study using LC-UV reported an LLOQ value of 50 ng/mL for withaferin A [56], whereas LC-MS/MS allowed an LLOQ of 0.2 ng/mL [44]. Conversely, LC-MS systems are more sophisticated and costly to purchase and maintain than LC-UV. Limited availability or access to this analytical technique may be a factor limiting the generation of bioanalytical data of withanolides alongside bioactivity of WS. Alternative, high-sensitivity methods that can be used for the detection of analytes in body fluids include immunoassays. Although such methods were not found for withanolides, immunoassays have been used for digoxin, a natural product with structural similarity to withanolides. However, a major disadvantage of immunoassays can be cross-reactivity with closely related compounds, i.e., limited selectivity as seen for digoxin [63].

High analytical sensitivity has been required, particularly for human studies where plasma withanolide C_max_ values have ranged from 0.09 ng/mL [57] to 49.50 ng/mL [58] but have typically been below 10 ng/mL (Table 3). However, only about half of the studies reported the use of fully validated LC-MS/MS methods, and of these, the lowest LLOQ achieved was 0.2 ng/mL for withaferin A [44]. Marney et al., 2024 compared LC-MS/MS (also known as LC–multiple reaction monitoring, LC-MRM-MS) of withanolides with LC coupled to high-resolution time of flight mass spectrometry methods operating in data-dependent acquisition mode (LC-HRMS/MS DDA) or in parallel-reaction monitoring mode with an inclusion list (LC-HRMS/MS PRM) [68]. The relative sensitivity of the three methods was found to vary by withanolide and species (molecular ion or adduct) measured, but LC-HRMS/MS PRM provided improved sensitivity over LC-MRM-MS for some withanolides (e.g., 12-deoxywithastramonolide, withanolide B). One challenge noted in this study was that for LC-MRM-MS, optimum source conditions (particularly temperature) differed for withanolide aglycones and glycosides. The use of an intermediate temperature may be necessary when both types of compounds are being analyzed in the same run.

Another challenge for studies of WS is that structurally distinct withanolides may be isobaric (i.e., have identical masses); for example, withaferin A, withanolide A, and 12-deoxywithastramonolide all have a monoisotopic mass of 470.3. Therefore, in experiments where more than one of these compounds may be present, it is essential that they are separated by the LC method to allow the compounds to be measured individually rather than together. In some studies where a complex extract was administered, these four compounds were separated and measured individually [49,50]. However, in others [51,52,53,57,58,59], these compounds were not all reported, making it unclear whether separation was achieved, allowing the MS/MS signal to be attributed to the stated isomer alone.

Four animal studies [40,45,50,55] applied these methods to examine tissue distribution of withanolides. Of particular relevance to the neuropsychiatric effects of WS [3] is the brain bioavailability of withanolides. Results varied but one study [50] reported the presence of withaferin A, withanolide A and B, withanone, and 12-deoxywithastramonolide in rat brain following oral administration of a WS extract. A limitation of these studies is that it was not reported if brain and other tissues were perfused (to remove residual blood) prior to measuring withanolide content.

Future animal studies should prioritize generating oral bioavailability data to better understand the impact of animal strain, vehicle, and product composition on the bioavailability of withanolides. However, the examples discussed here highlight how difficult it can be to compare studies, even when the same isolated compound is being administered, let alone evaluate their relevance to human use of WS. Given the large number of withanolides that have been identified in WS, there is also a need for researchers to explore the plasma PKs of lesser-known withanolides and their potential therapeutic roles. However, a major hurdle to achieving this goal is the limited number of commercially available, high-quality, isolated withanolides for use as reference samples. In addition, while withanolides have been the focus of research into WS’s active compounds, a variety of other phytochemical types have been isolated from WS extracts, including alkaloids, flavonoids, steroids, and nitrogen-containing compounds [69]. Pre-clinical studies suggest that non-withanolide compounds may be responsible for some of WS’s biological activity [70,71,72,73,74] and therefore, future studies should attempt to identify and quantify both withanolide and non-withanolide compounds in WS and explore potential synergistic effects between these compounds, particularly at biologically relevant concentrations. Finally, animal studies should further explore WS compound levels in biological matrices other than plasma; specifically, quantifying WS compound levels in the brain should be a priority given research suggesting a potential role for WS in neurodegenerative diseases and cognition [75,76,77,78].

For human studies, the most important research gap is the overall lack of human PK studies for WS. This issue is compounded by existing PK studies almost exclusively enrolling younger males; PK data are needed for female participants and for older participants. Connecting plasma PK data to clinical outcomes will require more rigorous PK studies; researchers should aim to (a) report more details of product preparation and formulation along with individual withanolide content to allow for easier comparisons across studies, (b) determine steady-state levels and safety after repeated dosing to better reflect real-world use, and (c) choose products and doses that have previously been administered in efficacy trials. Finally, researchers should attempt to quantify a greater variety of withanolides, especially those withanolides that have been detected in animal studies (e.g., withanone, withanolide B) that have not yet been analyzed in humans.

Two aspects of withanolide disposition in vivo that have received even less attention than their oral absorption, pharmacokinetics, and tissue distribution are the in vivo metabolism and excretion of these compounds. An in vitro study identified seven major metabolites of withaferin A in human and four in male rat microsomes [44] involving hydroxylation, hydrogenation, and hydrolysis, but no corresponding in vivo studies were found. In one report of withanolide metabolism in mice in vivo, orally administered withanoside IV (a glycoside, molecular weight 782.4) was not detected in plasma, whereas its aglycone sominone (*m/z* 459.2) and another, unidentified metabolite (*m/z* 475.3) were found [37]. It is notable that none of the PK studies reviewed here included treatment of plasma with glucuronidase or sulfatase enzymes. A comparison of plasma or urine analyte levels without or following enzyme treatment can be useful to detect the presence of Phase II conjugates, as has previously been described for other botanicals [61,62]. No reports were found regarding urinary or fecal levels of withanolides following their administration to pre-clinical models or humans. These are additional research gaps, complementary to pharmacokinetic studies, that need to be addressed.

## 5. Conclusions

WS is an Ayurvedic botanical that has garnered significant interest among researchers and consumers in recent years. Withanolides are among the main bioactive compounds found in WS. Determining the PKs and oral absorption of withanolides from WS is important for guiding future research of this promising botanical. This is an emerging field and validated bioanalytical methods using LC-MS/MS have been reported for the determination of a limited number of withanolides in animal and human plasma. Future research should focus on adapting these methods to address existing gaps in the field such as expanding the range of withanolides studied, exploring tissue distribution, metabolism, and excretion of these compounds, and studying the influence of product composition and biotransformation on their bioavailability.

## Figures and Tables

**Figure 1 nutrients-16-03836-f001:**
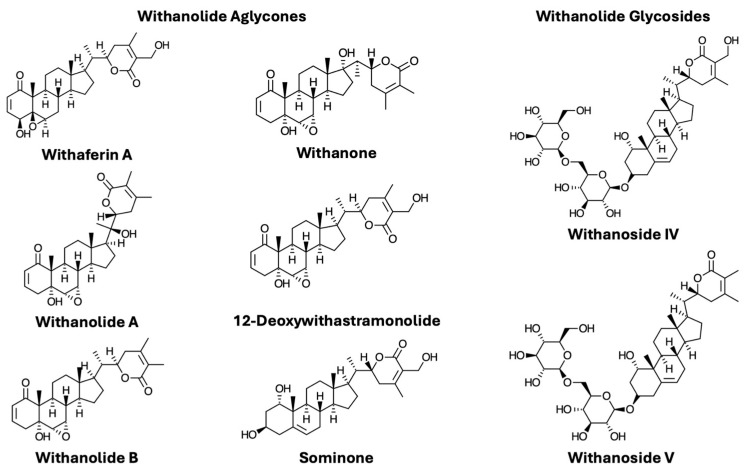
Chemical structures for withanolides analyzed in mammalian plasma samples.

**Table 1 nutrients-16-03836-t001:** Animal studies quantifying withanolides in plasma after administration of single withanolides.

Animal Model	Test Product(s)	Dosage, Route	Withanolide(s) Measured	Analysis Method	Plasma Pharmacokinetic Outcomes			Ref.
Male Swiss mice, haplotype H-2KdDb, six-to-eight-week-old	Withaferin A	50 mg/kg, IP	Withaferin A	HPLC ^✚^	Withaferin A			[40]
						Withaferin A was present in plasma at all measured time points (5 min, 15 min, 30 min, 1 h, 2 h, 4 h), reaching 10 µg/mL at 4 h); PK parameters NR.		
Female BALB/c mice, eight-to-ten-week-old	Withaferin A	70 mg/kg, PO 10 mg/kg, IV	Withaferin A	LC-MS/MS	Withaferin A			[41]
					PO	C_max_: 141.7 ± 16.8 ng/mL AUC_0-∞_: 436.1 ± 60.9 ng/mL·h T_max_: 0.5 (0.25–1.0) h T_1/2_: 2.7 ± 0.4 h C_max_: 3996.9 ± 557.6 ng/mL AUC_0-∞_: 3509.8 ± 302.4 ng/mL·h T_max_: NA T_1/2_: 0.6 ± 0.4 h		
					IV			
Male C57BL/6N mice, eight-week-old	Withaferin A	5 mg/kg, IP	Withaferin A	LC/MS	Withaferin A			[42]
						C_max_: 6.7 ± 0.9 ng/mL AUC_0-∞_: 86.6 ± 5.6 ng/mL·h T_max_: 20 min T_1/2_: 2.0 ± 0.6 h		
Female BALB/c mice, seven-to-eight-week-old	Withaferin A	4 mg/kg, IP	Withaferin A	LC-MS/MS	Withaferin A			[43]
						C_max_: 847.1 ng/mL AUC_0-∞_: NR T_max_: 0.083 h T_1/2_: 1.36 h		
Male Sprague Dawley rats	Withaferin A	10 mg/kg, PO 5 mg/kg, IV	Withaferin A	LC-MS/MS	Withaferin A			[44]
					PO	C_max_: 619 ± 125 ng/mL AUC_0-∞_: 2789 ± 683 ng/mL·h T_max_: 0.11 ± 0.07 h T_1/2_: 7.6 ± 3.3 h		
					IV	C_max_: 3048 ± 509 ng/mL AUC_0-∞_: 3685 ± 685 ng/mL·h T_max_: NA T_1/2_: 4.5 ± 1.1 h		
Sprague Dawley rats	Withaferin A	0.5, 1.5, 4.5 mg/kg, PO 4.5 mg/kg, IV	Withaferin A	LC-MS/MS	Withaferin A			[45]
					PO	0.5 mg/kg	C_max_: 6.05 ng/mL AUC_0-∞_: NR T_max_: 0.97 h T_1/2_: 1.00 h	
						1.5 mg/kg	C_max_: 14.60 ng/mL AUC_0-∞_: NR T_max_: 1.03 h T_1/2_: 0.78 h	
						4.5 mg/kg	C_max_: 21.80 ng/mL AUC_0-∞_: NR T_max_: 0.86 h T_1/2_: 1.15 h	
					IV	4.5 mg/kg	C_max_: 29.10 ng/mL AUC_0-∞_: NR T_max_: 0.33 h T_1/2_: 0.93 h	
Female Sprague Dawley rats	Withaferin A	4 mg/kg, IP 8 mg/kg, IP	Withaferin A	LC-MS/MS	Rats were treated with withaferin A five times per week for ten weeks. Blood was collected one hour after the last administration of withaferin A. Plasma levels of withaferin A at this timepoint were 65.93 ng/mL (4 mg/kg) and 138.1 ng/mL (8 mg/kg) per Dr. Singh’s lab (email communication, 2024).			[46]
Female BALB/c mice, estrogen-deficient bone loss model	Withaferin A	10 mg/kg, PO	Withaferin A	LC-UV	Withaferin A			[48]
						C_max_: 8410 ± 1400 ng/mL AUC_0-∞_: NR T_max_: Between 3 and 4 h T_1/2_: 7.1 ± 1.2 h		
Male Sprague Dawley rats	Withanolide A	25 mg/kg, PO 2 mg/kg, IV	Withanolide A	LC-MS/MS	Withanolide A			[47]
					PO	C_max_: 48.04 ± 5.78 ng/mL AUC_0-∞_: 76.41 ± 6.39 ng/mL·h T_max_: 0.33 ± 0.00 h T_1/2_: 2.23 ± 0.14 h		
					IV	C_max_: 85.53 ± 6.54 ng/mL AUC_0-∞_: 115.60 ± 17.54 ng/mL·h T_max_: 0.08 ± 0.00 h T_1/2_: 2.21 ± 0.21 h		
Male ddY mice, seven-week-old	Withanoside IV, isolated from a methanolic WS root extract	1000 µmol/kg (782.9 mg/kg), PO	Withanoside IV Sominone	HPLC-UV, LC/MS	Withanoside IV			[37]
						Not detected.		
					Sominone			
						Detected in the plasma beginning at 3 h, reaching maximal concentration at 7 h. Plasma concentrations NR.		

PO = oral administration, IV = intravenous administration, IP = intraperitoneal administration, PK = pharmacokinetic, NR = not reported, NA = not applicable. ^✚^ Detection method not reported.

**Table 2 nutrients-16-03836-t002:** Animal studies quantifying withanolides in plasma after administration of withanolide-containing plant extracts.

Animal Model	Test Product(s)	Dosage	Withanolide(s) Measured	Administered Dose of Withanolides ^$^	Analysis Method	Plasma Pharmacokinetic Outcomes		Ref.
Male Sprague Dawley rats	Alcoholic WS root extract	500 mg/kg, PO	Withaferin A Withanolide A Withanoside IV 12-DWS Withanoside V Withanolide B Withanone	Withaferin A: 4.84 mg/kg Withanolide A: 2.55 mg/kg Withanoside IV: 3.87 mg/kg 12-DWS: 1.51 mg/kg Withanoside V: 4.57 mg/kg Withanolide B: 0.793 mg/kg Withanone: 0.021 mg/kg	LC-MS/MS	Withaferin A		[49]
						C_max_: 124.42 ± 64.93 ng/mL AUC_0-∞_: 187.65 ± 20.49 ng/mL·h T_max_: 0.25 ± 0.00 h T_1/2_: 3.15 ± 0.61 h		
						Withanolide A		
						C_max_: 7.28 ± 3.34 ng/mL AUC_0-∞_: 7.53 ± 1.83 ng/mL·h T_max_: 0.33 ± 0.13 h T_1/2_: 0.73 ± 0.42 h		
						Withanoside IV		
						C_max_: 13.83 ± 3.73 ng/mL AUC_0-∞_: 22.94 ± 5.73 ng/mL·h T_max_: 0.75 ± 0.00 h T_1/2_: 1.10 ± 0.27 h		
						12-DWS		
						C_max_: 57.54 ± 7.52 ng/mL AUC_0-∞_: 92.25 ± 13.49 ng/mL·h T_max_: 0.29 ± 0.10 h T_1/2_: 1.73 ± 0.51 h		
						Withanoside V		
						Detected, but <LLOQ (3 ng/mL).		
						Withanolide B		
						Detected, but <LLOQ (3 ng/mL).		
						Withanone		
						Detected, but <LLOQ (3 ng/mL).		
Male Sprague Dawley rats	Withanolide-rich fraction (NMITLI-118R AF1) from a hydroethanolic (75:25) WS root extract	50 mg/kg, PO	Withaferin A Withanolide A 12-DWS Withanolide B Withanone	Withaferin A: 0.0015 mg/kg ^◆^ Withanolide A: 0.0895 mg/kg ^◆^ 12-DWS: 0.0010 mg/kg ^◆^ Withanolide B: 0.0315 mg/kg ^◆^ Withanone: 0.0165 mg/kg ^◆^	LC-MS/MS	Withaferin A		[50]
						C_max_: 6.50 ± 0.27 ng/mL AUC_0-∞_: 31.37 ± 2.23 ng/mL·h T_max_: 1.00 ± 0.00 h T_1/2_: 2.66 ± 0.24 h		
						Withanolide A		
						C_max_: 5.59 ± 0.34 ng/mL AUC_0-∞_: 18.71 ± 1.60 ng/mL·h T_max_: 1.00 ± 0.00 h T_1/2_: 1.89 ± 0.58 h		
						12-DWS		
						C_max_: 5.68 ± 0.39 ng/mL AUC_0-∞_: 15.14 ± 3.59 ng/mL·h T_max_: 1.00 ± 0.00 h T_1/2_: 2.08 ± 0.54 h		
						Withanolide B		
						C_max_: 6.45 ± 2.87 ng/mL AUC_0-∞_: 19.14 ± 5.41 ng/mL·h T_max_: 0.95 ± 0.11 h T_1/2_: 2.64 ± 1.18 h		
						Withanone		
						C_max_: 6.28 ± 0.41 ng/mL AUC_0-∞_: 16.88 ± 3.28 ng/mL·h T_max_: 0.95 ± 0.11 h T_1/2_: 2.04 ± 0.59 h		
Male Wistar rats	Polyherbal hydroalcoholic (40:60) extract (PHC3), containing (per 100 g): 15.4 g *Withania somnifera* 7.7 g *Hemidesmus indicus* 30.8 g *Emblica officinalis* 27 g *Aegle marmelos* 27 g *Ocimum sanctum*	200 mg/kg PHC3 extract, PO	Withaferin A	NR	LC-MS/MS	Withaferin A C_max_: 16.78 ± 5.32 ng/mL AUC_0-∞_: 1705 ± 28.87 ng/mL·h T_max_: 18 ± 0.12 min T_1/2_: 61.20 ± 9.87 min		[51]
Male Swiss albino adult mice	Hydroalcoholic (80:20) WS root extract (WSC) Withanolide aglycones- enriched fraction from WSC (WSAg)	200 mg/kg, PO	Withaferin A 12-DWS	WSC Withaferin A: 2.26 mg/kg 12-DWS: 5.36 mg/kg WSAg Withaferin A: 2.3 mg/kg 12-DWS: 141.12 mg/kg	LC-PDA	Withaferin A		[52]
						WSC	C_max_: 30 ng/mL AUC_0-∞_: 11.50 ng/mL·h T_max_: 1.32 h T_1/2_: 2.10 h	
						WSAg	C_max_: 60 ng/mL AUC_0-∞_: 11.73 ng/mL·h T_max_: 1.24 h T_1/2_: 2.97 h	
						12-DWS		
						WSC	C_max_: 120 ng/mL AUC_0-∞_: 2.55 ng/mL·h T_max_: 4.58 h T_1/2_: 6.61 h	
						WSAg	C_max_: 50 ng/mL AUC_0-∞_: 11.68 ng/mL·h T_max_: 1.73 h T_1/2_: 3.54 h	
Female Swiss albino mice	Aqueous WS root extract	1 g/kg, PO	Withaferin A Withanolide A	Withaferin A: 0.46 mg/kg Withanolide A: 0.48 mg/kg	LC-MS/MS	Withaferin A		[53]
						C_max_: 16.69 ± 4.02 ng/mL AUC_0-∞_: 1673.10 ± 54.53 ng/mL·h T_max_: 20 (20–30) min T_1/2_: 59.92 ± 15.90 min		
						Withanolide A		
						C_max_: 26.59 ± 4.47 ng/mL AUC_0-∞_: 2516.41 ± 212.10 ng/mL·h T_max_: 10 (10–30) min T_1/2_: 45.22 ± 9.95 min		
Male guinea pigs, normal group and psoriasis model	70% EtOH extract of dried flowers from *Datura metel* L.	5 g/kg *Datura metel* extract, PO	Withanolide B	Withanolide B: 19.83 mg/kg	LC-MS/MS	Withanolide B		[54]
						Normal	C_max_: 324.98 ± 43.39 ng/mL AUC_0-∞_: NR T_max_: 0.22 ± 0.043 h T_1/2_: 6.88 ± 1.95 h	
						Psoriasis	C_max_: 463.65 ± 41.46 ng/mL AUC_0-∞_: NR T_max_: 0.24 ± 0.034 h T_1/2_: 9.55 ± 4.00 h	

PO = oral administration, NR = not reported, LLOQ = lower limit of quantification, PDA = photo diode array, 12-DWS = 12-deoxywithastramonolide. **^$^** Calculated based on reported content of each withanolide in the test product administered. ^◆^ Calculated based on average withanolide content (%) from three batches of AF1 fraction.

**Table 3 nutrients-16-03836-t003:** Human studies quantifying withanolides in plasma.

Population	Test Product(s)	Dosage	Withanolide(s) Measured	Administered Dose of Withanolides	Analysis Method	Plasma Pharmacokinetic Outcomes		Ref.
Patients with advanced stage high-grade osteosarcoma (n = 13, age range: 13–43)	AshwaMAX 400: WS root extract, standardized to 4.5% withaferin A *w*/*w*; Pharmanza Herbal Pvt Ltd., Gujarat, India	Cohort 1: 1.6 g, PO Cohort 2: 2.4 g, PO Cohort 3: 3.2 g, PO Cohort 4: 4.8 g, PO	Withaferin A	Withaferin A: Cohort 1: 72 mg Cohort 2: 108 mg Cohort 3: 144 mg Cohort 4: 216 mg	LC-UV	Withaferin A Not detected in any sample. LLOQ = 50 ng/mL		[56]
Healthy adult males (n = 14, age range: 23–42 years)	Prolanza™: WS root extract (type of extraction NR), sustained release 300 mg capsules, standardized to 15 mg withanolides; Inventia healthcare Ltd. and Laila nutraceuticals, India KSM-66^®^: Aqueous WS root extract, capsules standardized to 15 mg withanolides; Shri Kartikeya Pharma, India;	Prolanza: 2 caps, PO KSM-66: 2 caps, PO	Withanolide A 12-DWS	Prolanza: 30 mg withanolides KSM-66: 30 mg withanolides	LC-MS/MS	Withanolide A		[57]
						Prolanza	C_max_: 0.49 ± 0.34 ng/mL AUC_0-∞_: NR T_max_: 1 h T_1/2_: 7.46 ± 5.92 h	
						KSM-66	C_max_: 0.09 ± 0.10 ng/mL AUC_0-∞_: NR T_max_: 1 h T_1/2_: 0.74 ± NE h	
						12-DWS		
						Prolanza	C_max_: 2.67 ± 1.04 ng/mL AUC_0-∞_: NR T_max_: 2 h T_1/2_: 7.53 ± 2.67 h	
						KSM-66	C_max_: 0.61 ± 0.31 ng/mL AUC_0-∞_: NR T_max_: 2 h T_1/2_: 2.29 ± 0.43 h	
Healthy adult males (n = 16, mean age: 33.8 years)	WS-35: Hydroalcoholic WS root and leaf extract, standardized to 40% total withanolides comprising 35% withanolide glycosides; Arjuna Natural Pvt Ltd., India WS-2.5: Type of extract NR, standardized to 2.5% withanolides; Natura Biotechnol, India	WS-35: 480 mg, PO Single dose WS-2.5: 7400 mg, PO Single dose	Withaferin A Withanolide A Withanoside IV	WS-35: 185 mg withanolides WS-2.5: 185 mg withanolides	LC-MRM/MS	Withaferin A		[58]
						WS-35	C_max_: 49.50 ± 1.24 ng/mL AUC_0-∞_: 748.95 ± 23.90 ng/mL·h T_max_: 2.28 ± 0.09 h T_1/2_: 10.35 ± 0.47 h	
						WS-2.5	C_max_: 10.79 ± 0.13 ng/mL AUC_0-∞_: 44.76 ± 0.57 ng/mL·h T_max_: 1.5 ± 0 h T_1/2_: 1.88 ± 0.03 h	
						Withanolide A		
						WS-35	C_max_: 4.74 ± 0.22 ng/mL AUC_0-∞_: 69.23 ± 15.42 ng/mL·h T_max_: 1.83 ± 0.06 h T_1/2_: 10.95 ± 2.20 h	
						WS-2.5	C_max_: 2.93 ± 0.06 ng/mL AUC_0-∞_: 17.11 ± 0.57 ng/mL·h T_max_: 2.20 ± 0.08 h T_1/2_: 4.03 ± 0.27 h	
						Withanoside IV		
						WS-35	C_max_: 7.23 ± 0.42 ng/mL AUC_0-∞_: 92.52 ± 14.34 ng/mL·h T_max_: 1.76 ± 0.07 h T_1/2_: 8.86 ± 1.15 h	
						WS-2.5	C_max_: 2.67 ± 0.04 ng/mL AUC_0-∞_: 11.35 ± 0.35 ng/mL·h T_max_: 1.57 ± 0.07 h T_1/2_: 2.43 ± 0.18 h	
Healthy adult males (n = 18, mean age: 29.4 years)	Witholytin^®^: WS root extract, standardized to ≥1.5% *w/w* of total withanolides; Verdure Science, Noblesville, IN, USA	1 cap, PO Single dose	Withaferin A Withanolide A 12-DWS Withanoside IV Withanoside V Sominone	Per capsule: 7.97 mg total withanolides 2.42 mg withanoside IV 1.89 mg withanoside V Concentrations of other withanolides NR	LC-MS/MS	Withaferin A		[59]
						C_max_: 2.88 ± 0.98 ng/mL AUC_0-∞_: 7.81 ± 2.59 ng/mL·h T_max_: 0.90 ± 0.27 h T_1/2_: 4.00 ± 1.80 h		
						Withanolide A		
						C_max_: 2.93 ± 1.32 ng/mL AUC_0-∞_: 15.93 ± 7.48 ng/mL·h T_max_: 1.36 ± 0.85 h T_1/2_: 4.19 ± 1.09 h		
						12-DWS		
						C_max_: 5.50 ± 2.00 ng/mL AUC_0-∞_: 24.26 ± 12.02 ng/mL·h T_max_: 1.38 ± 0.52 h T_1/2_: 2.73 ± 0.54 h		
						Withanoside IV		
						C_max_: 0.64 ± 0.21 ng/mL AUC_0-∞_: 4.86 ± 1.88 ng/mL·h T_max_: 1.64 ± 0.99 h T_1/2_: 4.41 ± 1.54 h Withanoside V Detected, but <LLOQ (0.25 ng/mL). Sominone Detected, but <LLOQ (0.25 ng/mL).		

PO = oral administration, 12-DWS = 12-deoxywithastramonolide, NR = not reported, LLOQ = lower limit of quantification, NE = statistical analysis not possible.

**Table 4 nutrients-16-03836-t004:** Methods used for detecting withanolides in animal plasma.

Method	Validation	Species	Plasma Preparation Volume Used; Method	LC-MS Method (and UV Wavelength if Applicable)	Compounds and MRM Transition Used		Ref.
					Withanolides	Internal Standard	
LC-MS/MS	Yes	Rat	85 µL; solid phase extraction using Bond Elute C18 cartridges	Column: ReproSil Gold 100C18-XBD, 50 × 4.6 mm; 1.8 µm Mobile Phase: Aqueous formic acid (0.1%) (A) and acetonitrile (B) MRM Mode: Positive ionization mode	Withanone (417.25/263.15) Withaferin A (471.25/67.05) Withanolide A (488.3/471.25) Withanolide B (472.30/109.15) 12-DWS (471.25/67.05) Withanoside V (784.45/443.3) Withanoside IV (800.45/459.3)	Fluoxymesterone (337.2/91.15) Difenoconazole (406.1/336.9)	[49]
LC-MS/MS	Yes	Rat	200 µL; protein precipitation with 4% sulfosalicylic acid followed by extraction into 100% ethyl acetate	Column: A Kinetex^®^ 1.7 µm C18 100 Å (100 × 3 mm, S/No. H20–111310, Batch No. XD-4475-YO) Mobile Phase: Acetonitrile (A) and 10 mM ammonium acetate (B) in milli-Q water (60:40 *v*/*v*). MRM Mode: Positive ionization mode	Withaferin A (471.4/281.2)	Fluoxymesterone (337.2/91.1)	[41]
LC-MS	No	Rat	100 µL; protein precipitation with acetonitrile	Column: C18—250 mm × 4.6 mm and 5 µm Mobile Phase: Water and acetonitrile (60:40 *v*/*v*) MRM Mode: Positive and negative ionization modes in a single run	Withanolide A (NR)	NR	[51]
LC-MS	No	Mouse	Volume NR; protein precipitation with acetonitrile	NR	Withaferin A (NR)	NR	[42]
LC-MS/MS	No	Mouse	Volume NR; protein precipitation followed by solvent evaporation	Column: NR Mobile Phase: NR MRM Mode: Positive ionization mode	Withaferin A (NR)	NR	[43]
LC-MS/MS	Yes	Rat	50 µL; protein precipitation with acetonitrile followed by extraction into TBME	Column: Phenomenex Luna (5 µm, C18, 150 × 4.60 mm) Mobile Phase: Acetonitrile and 0.1% formic acid in water (95:05 *v*/*v*) MRM Mode: Positive ionization mode	Withanone (488.300/263.200) Withaferin A (471.211/299.210) Withanolide A (471.246/263.190) Withanolide B (455.305/109.15) 12-DWS (471.254/263.172)	Phenacetin (180.200/110.200)	[50]
LC-MS/MS	Yes	Rat	100 µL; protein precipitation with methanol and internal standard solution followed by extraction into ethyl acetate	Column: Venusil MP C18 column (50 × 2.1 mm, 5 μm) Mobile Phase: Acetonitrile and water at ratio of 5/95 *v*/*v* (A), and acetonitrile and water, 95/5 *v*/*v* (B). Both phases contained 10 mM ammonium acetate MRM Mode: Positive ionization mode	Withaferin A (471.4/281.2)	Alisol C 23-acetate (529.6/451.5)	[44]
LC-MS/MS	Yes	Guinea pig	100 µL; protein precipitation with 90% methanol and IS solution	Column: Acquity UPLC BEH C18 column (2.1 × 100 mm^2^, 1.7 μm) Mobile Phase: Aqueous 0.1% formic acid (A) and acetonitrile with 0.1% formic acid (B) MRM Mode: Positive ionization mode	Withanolide B (455.1/109.4)	Obakunone (455.1/161.1)	[54]
LC-MS/MS	Yes	Mouse	100 µL; plasma mixed with IS solution (IS prepared with diH2O) and then extracted with TBME	Column: Reversed-phase Hypurity C18 column (50 mm × 4.6 mm, 5 μm; Thermo scientific, Mumbai, India) Mobile Phase: Methanol and 2 mM ammonium acetate (95:5, *v*/*v*) MRM Mode: Positive ionization mode	Withaferin A (471.3/281.2) Withanolide A (488.3/263.1)	Tianeptine (437.2/292.2) Clonazepam (315.9/270)	[53]
LC-MS/MS	No	Rat	100 µL; protein precipitation with acetonitrile, 10 mM ammonium formate buffer (0.1% formic acid), and 5 mM NaHCO3 solution followed by extraction into TBME	Column: Phenomenex Luna C18 column (4.6 × 250 mm, 5.0 μm) Mobile Phase: Acetonitrile and methanol and 10 mM ammonium formate buffer with 0.1% formic acid (50:20:30, %*v*/*v*/*v*) MRM Mode: Positive ionization mode	Withanolide A (471.22/ 263.2)	Carbamazepine (237.08/194.1)	[47]
LC-UV	No	Mouse	NR	Column: Stainless steel C18 column, 5 μm ODS2, 4.5 × 250 Mobile Phase: Acetonitrile and 0.2% orthophosphoric acid (75:25, *v*/*v*) MS Mode: NR UV wavelength: 290 nm	Withaferin A (NR)	NR	[40]
LC-UV, LC-MS	No	Mouse	300 µL; solid phase extraction using Oasis HLB 1 cc Extraction Cartridges	Column: LV-UV Symmetry Shield RP18 column (5 μm, 4.6 × 250 mm) LC/MS–Imtakt Cadenza CD-C18 column (3 μm, 2.0 × 150 mm) Mobile Phase: Water (A) and acetonitrile (B) differing gradients used for LC-UV and LC-MS MS Mode: Positive ionization mode UV wavelength: 220 nm	Withanoside IV (NR) Sominone (*m/z* 459)	NR	[37]
LC-MS/MS	Yes	Rat	100 µL; saline solution (10% ascorbic acid) added to samples followed by IS and extraction into ethylacetate	Column: Hypurity C18 (50 × 4.6 mm, 5 μm; Thermo scientific) Mobile Phase: Acetonitrile and water (35:65, *v*/*v*) MRM Mode: Positive ionization mode	Withaferin A (471.1/281)	Withanolide A (488.1/263)	[45]
LC-UV	No	Mouse	Volume NR; method NR	Column: LiChrospher^®^ LiChroCART^®^ C18 column (250 mm, 4 mm, 5 mm) Mobile Phase: Potassium dihyrdogren phosphate buffer (0.05M) and triethyl amine (0.1%) pH- 2.5 and acetonitrile (65:35) MRM Mode: NA UV wavelength: 370 nm	Withaferin A (NR)	NR	[48]

MRM = Multiple Reaction Monitoring, 12-DWS = 12-deoxywithastramonolide, NA = not applicable. NR = Not reported, IS = internal standard. Analyte monoisotopic masses: 12-DWS, 470.3; sominone, 458.3; withaferin A, 470.3; withanolide A, 470.3; withanolide B, 454.3; withanone, 470.3; withanoside IV, 782.4; withanoside V, 766.4.

**Table 5 nutrients-16-03836-t005:** Methods used for detecting withanolides in human plasma.

Method	Validation	Species	Plasma Preparation Volume Used; Method	LC-MS Method (and UV Wavelength if Applicable)	Compounds and MRM Transition Used		Ref.
					Withanolides	Internal Standard	
LC-MS/MS	No	Human	85 µL; solid phase extraction using Agilent, Bond Elute PLEXA Cartridges	Column: Exsil Mono 100 C18, 3 µm, (100 × 4.6) mm Mobile Phase: Acetonitrile and 10 mM ammonium formate in water (70:30) with 0.1% glacial acetic acid MRM mode: Positive ionization mode	Withanolide A (488.5/263.2) 12-DWS (471.4/263.2)	Atorvastatin D5 (564.4/445.4)	[57]
LC-MS/MS	Yes	Human	380 µL; solid phase extraction using Bond Elute C18 SPE cartridges	Column: Agilent ZORBAX Eclipse Plus (4.6 × 100 mm, 3.5 µm) C18 column Mobile Phase: 1 mm ammonium formate in water (A) and acetonitrile (B) MRM mode: Positive ionization mode	Withaferin A (471.3/281.1) Withanolide A (488.3/471.2) Withanoside V (784.45/443.3) Withanoside IV (800.45/459.3) 12-DWS (488.3/471.2)	Fluoxymesterone (337.2/91)	[59]
LC-MS/MS	Yes	Human	Volume NR; solid phase extraction using OasisR HLB Cartridges	Column: Acquity UPLC BEH phenyl C18 column 100 × 2.1 mm L.D., 1.7 μm Mobile Phase: Formic acid 0.1% in water (A) and in acetonitrile (B) MRM Mode: Positive and negative ESI (negative ion mode for withanoside IV and positive ion mode for withaferin A, withanolide A, and internal standard)	Withaferin A (471.1711/94.95) Withanolide A (471.185/263.1045) Withanoside IV (827.4404/763.3592)	Taineptine (437.0904/292.0991)	[58]
LC-UV	No	Human	Volume NR; simple Protein Precipitation (details NR)	Column: Reversed-phase C18 column Mobile Phase: Water (A) and acetonitrile (B). UV wavelength: 225 nm.	Withaferin A (NR)	NR	[56]

MRM = Multiple Reaction Monitoring, 12-DWS = 12-deoxywithastramonolide, NR = not reported. Analyte monoisotopic masses: 12-DWS, 470.3; withaferin A, 470.3; withanolide A, 470.3; withanoside IV, 782.4; withanoside V, 766.4.

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
