# Peer review of "Quantifying Withanolides in Plasma: Pharmacokinetic Studies and Analytical Methods"

_nutrients, 2024, doi:10.3390/nu16223836_

Round 1

Reviewer 1 Report

Comments and Suggestions for Authors

The Authors described in detail the information collected on withanolides in plasma. However, some important information is missing from the text. The following is a list of comments.

The introduction lacks information on adverse effects that may occur after the use of WS

Lines 149-150 - the Authors mention WS assay conducted in biological material other than plasma. It would have been useful to at least briefly describe how this biological material was prepared, since Cmax, among others, were given

Lines 247-249 - when describing the SPE process, it would be worthwhile to add information on the solvents used during sample preparation and whether the cartridge was washed once or repeatedly during SPE.

Line 267-268 - anti-cancer activity is a very broad term. Against which exact cancers were the described compounds used.

The analytical methods summary should also include information on the size of the biological sample used

Author Response

We thank the reviewer for their careful appraisal of our submission. We have addressed their comments as described below and are grateful for this opportunity to greatly improve our manuscript based on their recommendations. While doing additional research to address all the reviewers’ comments, an additional PK study (Khedgikar et al. 2013) was found and has been added to the manuscript where appropriate. Changes to the manuscript have been highlighted in yellow.

Comment 1: The introduction lacks information on adverse effects that may occur after the use of WS

Response 1: We have added WS safety information (lines 40-54) to the introduction, covering common and rare side effects, patient populations that should avoid WS, and the potential for herb/drug interactions. Toxicity data for individual withanolides and the relationship to PK data is addressed in the discussion (lines 516-545).

Comment 2: Lines 149-150 - the Authors mention WS assay conducted in biological material other than plasma. It would have been useful to at least briefly describe how this biological material was prepared, since Cmax, among others, were given

Response 2: We’ve added a paragraph (lines 295 – 305) describing how the tissues  homogenates were prepared a in those studies.

Comment 3: Lines 247-249 - when describing the SPE process, it would be worthwhile to add information on the solvents used during sample preparation and whether the cartridge was washed once or repeatedly during SPE.

Response 3: We’ve added a couple of sentences describing additional details of the SPE processes used (lines 277-281). We respectfully submit that detailed information (e.g., the number of cartridge washes used in each method) is outside the scope of this general review. Readers interested in knowing these details for individual methods are recommended to obtain the original papers.

Comment 4: Line 267-268 - anti-cancer activity is a very broad term. Against which exact cancers were the described compounds used.

Response 4: We agree and have added more details to that sentence in the introduction, listing out the cancer types and the mechanisms of action that have been observed (lines 65-68). This section has been moved to the introduction per another reviewer’s request.

Comment 5: The analytical methods summary should also include information on the size of the biological sample used

Response 5: The volume of plasma used for each method (if reported) has now been included in Tables 4 and 5. We agree that this information will be useful to readers when comparing methods.

Reviewer 2 Report

Comments and Suggestions for Authors

Speers and colleagues present a comprehensive summary of the current state of research on the plasma pharmacokinetics of withanolides in Withania somnifera.

Although the review mentions the importance of studying biotransformation and product composition in relation to bioavailability, it doesn't provide detailed insights into how these factors could influence absorption and effectiveness. It would have been useful to expand on existing studies or propose mechanisms through which different formulations affect pharmacokinetics.

The review notes a disproportionate number of animal studies (fifteen) compared to human studies (four). This highlights a significant gap in the literature, and the review should stress the need for more human pharmacokinetic studies to better understand the therapeutic relevance of these findings.

There is no in-depth comparison of the studies' methodologies, which could provide insights into the variability in withanolide plasma levels. Factors like dosage, route of administration, and formulation are noted but not systematically analyzed. More cross-study comparisons would help identify trends or inconsistencies in the data.

The review briefly mentions reversed-phase liquid chromatography coupled to mass spectrometry (LC-MS/MS) as a key analytical method but lacks critical analysis of the limitations and potential improvements of these techniques. It would benefit from a discussion on the sensitivity, specificity, and reliability of these methods, especially in detecting low plasma concentrations of withanolides in humans.

The review primarily focuses on a few well-known withanolides like withaferin A and withanolide A but doesn’t explore lesser-known or newly discovered withanolides. Given the complexity of WS, further studies on a broader range of withanolides could unveil additional therapeutic potentials.

Although the review emphasizes the importance of plasma pharmacokinetics, it doesn’t adequately link this information to clinical outcomes. A discussion on how specific plasma concentrations relate to therapeutic efficacy or toxicity would enhance the clinical relevance of the findings.

Author Response

We thank the reviewer for their careful appraisal of our submission. We have addressed their comments as described below and are grateful for this opportunity to greatly improve our manuscript based on their recommendations. While doing additional research to address all the reviewers’ comments, an additional PK study (Khedgikar et al. 2013) was found and has been added to the manuscript where appropriate. Changes to the manuscript have been highlighted in yellow.

Comment 1: Although the review mentions the importance of studying biotransformation and product composition in relation to bioavailability, it doesn't provide detailed insights into how these factors could influence absorption and effectiveness. It would have been useful to expand on existing studies or propose mechanisms through which different formulations affect pharmacokinetics.

Response 1: We have provided some possible reasons underlying the observed variations in bioavailability of individual withanolides in animal models in relation to biotransformation (lines 330 to 370), as well as other factors like sex of animal (384-390, 400-406), other compounds in the administered product (424-456) and also in humans (lines 468-480). The influence of product composition is specifically addressed in lines 436-456 for animal models and lines 468-480 in the human studies. As we describe in lines 626-640, currently, little is known about the intestinal or systemic biotransformation of withanolides. Therefore, we are only able to discuss hypothetical mechanisms underlying the observed differences.

Comment 2: The review notes a disproportionate number of animal studies (fifteen) compared to human studies (four). This highlights a significant gap in the literature, and the review should stress the need for more human pharmacokinetic studies to better understand the therapeutic relevance of these findings.

Response 2: In the discussion, we’ve added language emphasizing how the lack of human studies is the most significant research gap and included recommendations to increase the rigor of future studies with the goal of connecting PK data to clinical outcomes (lines 615-623).

Comment 3: There is no in-depth comparison of the studies' methodologies, which could provide insights into the variability in withanolide plasma levels. Factors like dosage, route of administration, and formulation are noted but not systematically analyzed. More cross-study comparisons would help identify trends or inconsistencies in the data.

Response 3: We agree that a detailed analysis could help understand the factors influencing withanolide plasma levels. A greatly expanded section comparing studies and reasons underlying the similarities and differences seen is now given in the discussion lines 330-485.

Comment 4: The review briefly mentions reversed-phase liquid chromatography coupled to mass spectrometry (LC-MS/MS) as a key analytical method but lacks critical analysis of the limitations and potential improvements of these techniques. It would benefit from a discussion on the sensitivity, specificity, and reliability of these methods, especially in detecting low plasma concentrations of withanolides in humans.

Response 4: We now include a section elaborating on the advantages, limitations (lines 551-562) and potential improvements to the use of LC-MS/MS for plasma analysis (lines 567-578).

Comment 5: The review primarily focuses on a few well-known withanolides like withaferin A and withanolide A but doesn’t explore lesser-known or newly discovered withanolides. Given the complexity of WS, further studies on a broader range of withanolides could unveil additional therapeutic potentials.

Response 5: We have added a discussion on analyzing other withanolides and non-withanolide compounds in WS to the discussion section on research gaps (lines 599-610).

Comment 6: Although the review emphasizes the importance of plasma pharmacokinetics, it doesn’t adequately link this information to clinical outcomes. A discussion on how specific plasma concentrations relate to therapeutic efficacy or toxicity would enhance the clinical relevance of the findings.

Response 6: We have now included several sections in the discussion describing the importance of being able to link plasma PK parameters to clinical outcomes (lines 486-504, 538-545, 615-623), as well as safety and toxicity (lines 505-537), and describe the limitations of being able to do this adequately with currently available data.

Reviewer 3 Report

Comments and Suggestions for Authors

With this narrative review, the authors present a well-written and well-structured survey the pharmacokinetics of withanolides from Withania somnifera (L.) Dunal. Ample information is provided about various pharmacokinetic parameters in different animals (rodents) as well as in humans. The discussion of the data from the literature is clear and contributes to the understanding of current knowledge as well as of gaps and ‘missing links’. Also, applied analytical procedures are presented in detail and discussed.

I have a few suggestions to consider, that might be useful to further improve the paper.

1. Pay some attention to the chemistry of withanolides and present structural formulas of the most abundant and principal ones. This will illustrate the variations (also aglucones versus glycosides, see next point). 

2. Both aglucones and glycosides of withanolides are present in the plant material. What differences are seen (or expected) in bioavailability, the rate of absorption, pharmacokinetics in general, when aglucones and glycosides are compared? 

3. Is anything known about metabolites of withanolides measured in blood or serum?

4. Could the authors make any recommendations for preparations of W. somnifera, for instance, regarding standardization (probably also with respect to raw materials and their quality) and regarding the formulation for oral use (see, e.g., lines 297, 333)?

5. Does the literature report any adverse effects possibly caused by too high plasma levels?

Author Response

We thank the reviewer for their careful appraisal of our submission. We have addressed their comments as described below and are grateful for this opportunity to greatly improve our manuscript based on their recommendations. While doing additional research to address all the reviewers’ comments, an additional PK study (Khedgikar et al. 2013) was found and has been added to the manuscript where appropriate. Changes to the manuscript have been highlighted in yellow.

Comment 1: Pay some attention to the chemistry of withanolides and present structural formulas of the most abundant and principal ones. This will illustrate the variations (also aglucones versus glycosides, see next point). 

Response 1: We’ve added Figure 1 to the introduction, which shows the chemical structures for the withanolides discussed in the review, separated into aglycones and glycosides (line 62-63).

Comment 2: Both aglucones and glycosides of withanolides are present in the plant material. What differences are seen (or expected) in bioavailability, the rate of absorption, pharmacokinetics in general, when aglucones and glycosides are compared? 

Response 2:  There is limited data on PK parameters of withanolide glycosides, but we now include a section in the discussion outlining what is known and comparing existing data on PK parameters of glycosides and aglycones (lines 330-358).

Comment 3: Is anything known about metabolites of withanolides measured in blood or serum?

Response 3:  There is limited data about withanolide metabolism. A paragraph highlighting what is currently known and the nature of research gaps is now included at the end of the discussion (lines 626-640).

Comment 4: Could the authors make any recommendations for preparations of W. somnifera, for instance, regarding standardization (probably also with respect to raw materials and their quality) and regarding the formulation for oral use (see, e.g., lines 297, 333)?

Response 4:  We agree that it would be helpful to be able to provide guidance of this type to readers. However, due to the variation of PK parameters with formulation (discussed in lines 468-485), limited information on the composition of the commercial products described, and most importantly, limited data linking plasma levels to therapeutic activity or adverse effects, we are unable to make any specific recommendations. We’ve included additional research gaps to address in future human trials that would allow for such recommendations to be made in lines 618-623.

Comment 5: Does the literature report any adverse effects possibly caused by too high plasma levels?

Response 5: While we’ve added sections on WS adverse events in humans (lines 40-54) and pre-clinical studies on withanolide toxicity (lines 505-545), there is no data linking specific plasma levels of individual withanolides to negative effects as we now discuss (lines 516-535). The only data semi-related to this question is the study linking a specific withanolide (withanone) to the potential hepatotoxic effects of WS, but this was an in vitro study (lines 533-537).

Reviewer 4 Report

Comments and Suggestions for Authors

The manuscript titled "The quantification of withanolides in plasma: a narrative review" by Alex Speers et al. provides a comprehensive summary of the current knowledge regarding the plasma pharmacokinetics of withanolides found in Withania somnifera (ashwagandha) and the analytical methods developed to detect them in plasma. The review is timely and relevant given the increasing popularity of ashwagandha as a botanical dietary supplement. Here are my comments and suggestions for improvement:

1. The manuscript is well-structured and clearly written. The separation of animal and human studies allows for easy comparison and understanding of the current state of research in both areas. However, the manuscript could benefit from a more detailed introduction that outlines the significance of withanolides, their purported health benefits, and the rationale behind the need for this review.

2. The presentation of pharmacokinetic data is thorough and well-organized. It would be beneficial to include a discussion on the variability observed in plasma concentrations of withanolides across different studies. Factors such as dosage, route of administration, and formulation could be explored to understand these variations.

3. While the manuscript focuses on the quantification of withanolides, it would be valuable to include a brief section on the safety and toxicity profiles of the main withanolides discussed. This could provide a more comprehensive view of their therapeutic potential.

4. The authors mention the need for future research to address existing gaps in the field. It would be helpful to expand on this section, suggesting specific areas for future research, such as the investigation of withanolides in other biological matrices, the exploration of synergistic effects with other compounds, or the study of long-term pharmacokinetics and safety.

Author Response

We thank the reviewer for their careful appraisal of our submission. We have addressed their comments as described below and are grateful for this opportunity to greatly improve our manuscript based on their recommendations. While doing additional research to address all the reviewers’ comments, an additional PK study (Khedgikar et al. 2013) was found and has been added to the manuscript where appropriate. Changes to the manuscript have been highlighted in yellow.

Comment 1: The manuscript is well-structured and clearly written. The separation of animal and human studies allows for easy comparison and understanding of the current state of research in both areas. However, the manuscript could benefit from a more detailed introduction that outlines the significance of withanolides, their purported health benefits, and the rationale behind the need for this review.

Response 1: We have moved information about the potential therapeutic benefits of withanolides from the discussion to the introduction and included additional details (lines 65-74). In addition, we expanded the paragraph describing our rationale for this review (lines 75-84). Per another reviewer’s suggestion, we’ve also added a figure of the chemical structures for the withanolides discussed in this review (lines 62-63).

Comment 2: The presentation of pharmacokinetic data is thorough and well-organized. It would be beneficial to include a discussion on the variability observed in plasma concentrations of withanolides across different studies. Factors such as dosage, route of administration, and formulation could be explored to understand these variations.

Response 2:  We have discussed several reasons underlying the observed variations in bioavailability of individual withanolides in animal models (lines 330 to 370, 384-390, 400-406, 424-456) and in humans (lines 468-480). The influence of product composition is specifically addressed in lines 436-456 for animal models and lines 468-480 in the human studies.  

Comment 3: While the manuscript focuses on the quantification of withanolides, it would be valuable to include a brief section on the safety and toxicity profiles of the main withanolides discussed. This could provide a more comprehensive view of their therapeutic potential.

Response 3: We have added a section on withanolide toxicity to the discussion (lines 516-545). We have also added general safety information regarding WS to the introduction (lines 40-54).

Comment 4: The authors mention the need for future research to address existing gaps in the field. It would be helpful to expand on this section, suggesting specific areas for future research, such as the investigation of withanolides in other biological matrices, the exploration of synergistic effects with other compounds, or the study of long-term pharmacokinetics and safety.

Response 4: We have expanded our discussion on research gaps significantly (lines 599-623), addressing several of the topics mentioned in your comment as well as additional recommendations for future research.

Round 2

Reviewer 2 Report

Comments and Suggestions for Authors

After seeing the major modifications made by the authors in the manuscript, I believe it can be published.